# Do Image Classifiers Generalize Across Time?

## Abstract

We study the robustness of image classifiers to temporal perturbations derived from videos. As part of this study, we construct `ImageNet-Vid-Robust` and `YTBB-Robust`, containing a total 57,897 images grouped into 3,139 sets of perceptually similar images. Our datasets were derived from ImageNet-Vid and Youtube-BB respectively and thoroughly re-annotated by human experts for image similarity. We evaluate a diverse array of classifiers pre-trained on ImageNet and show a median classification accuracy drop of 16 and 10 percent on our two datasets. Additionally, we evaluate three detection models and show that natural perturbations induce both classification as well as localization errors, leading to a median drop in detection mAP of 14 points. Our analysis demonstrates that perturbations occurring naturally in videos pose a substantial and realistic challenge to deploying convolutional neural networks in environments that require both reliable and low-latency predictions.

## 1 Introduction

Convolutional neural networks (CNNs) still exhibit many troubling failure modes. At one extreme, $\ell_p$-adversarial examples cause large drops in accuracy for state-of-the-art models while relying only on visually imperceptible changes to the input image (Goodfellow et al., 2014; Biggio and Roli, 2018). However, this failure mode usually does not pose a problem outside a fully adversarial context because carefully crafted $\ell_p$-perturbations are unlikely to occur naturally in the real world.

To study more realistic failure modes, researchers have investigated benign image perturbations such as rotations & translations, colorspace changes, and various image corruptions (Fawzi and Frossard, 2015; Engstrom et al., 2017; Fawzi and Frossard, 2015; Hendrycks and Dietterich, 2019). However, it is still unclear whether these perturbations reflect the robustness challenges arising in real data since the perturbations also rely on synthetic image modifications.

Recent work has therefore turned to videos as a source of *naturally occurring* perturbations of images (Zheng et al., 2016; Azulay and Weiss, 2018; Gu et al., 2019). In contrast to other failure modes, the perturbed images are taken from existing image data without further modifications that make the task more difficult. As a result, robustness to such perturbations directly corresponds to performance improvements on real data.

However, it is currently unclear to what extent such video perturbations pose a significant robustness challenge. Azulay and Weiss (2018) and Zheng et al. (2016) only provide anecdotal evidence from a small number of videos. Gu et al. (2019) go beyond individual videos and utilize a large video dataset (Real et al., 2017) in order to measure the effect of video perturbations more quantitatively. In their evaluation, the best image classifiers lose about 3% accuracy for video frames up to 0.3 seconds away. However, the authors did not employ humans to review the frames in their videos. Hence the accuracy drop could also be caused by significant changes in the video frames (e.g., due to fast camera or object motion). Since the 3% accuracy drop is small to begin with, it remains unclear whether video perturbations are a robustness challenge for current image classifiers.

We address these issues by conducting a thorough evaluation of robustness to natural perturbations arising in videos. As a cornerstone of our investigation, we introduce two test sets for evaluating model robustness: `ImageNet-Vid-Robust` and `YTBB-Robust`, carefully curated from the ImageNet-Vid and Youtube-BB datasets, respectively (Russakovsky et al., 2015; Real et al., 2017). All images in the two datasets were screened by a set of expert labelers to ensure high annotation quality and minimize selection biases that arise when filtering a dataset with CNNs. To the best of

our knowledge these are the first datasets of their kind, containing tens of thousands of images that are *human reviewed* and grouped into thousands of perceptually similar sets. In total, our datasets contain 3,139 sets of temporally adjacent and visually similar images (57,897 images total).

We then utilize these datasets to measure the accuracy of current CNNs to small, naturally occurring perturbations. Our testbed contains over 45 different models, varying both architecture and training methodology (adversarial training, data augmentation, etc.). To better understand the drop in accuracy due to natural perturbations, we also introduce a robustness metric that is more stringent than those employed in prior work. Under this metric, we find that natural perturbations from `ImageNet-Vid-Robust` and `YTBB-Robust` induce a median accuracy drop of 16% and 10% respectively for classification tasks and a median 14 point drop in mAP for detection tasks.[1] Even for the best-performing classification models, we observe an accuracy drop of 14% for `ImageNet-Vid-Robust` and 8% for `YTBB-Robust`.

Our results show that robustness to natural perturbations in videos is indeed a significant challenge for current CNNs. As these models are increasingly deployed in safety-critical environments that require both high accuracy and low latency (e.g., autonomous vehicles), ensuring reliable predictions on *every frame* of a video is an important direction for future work.

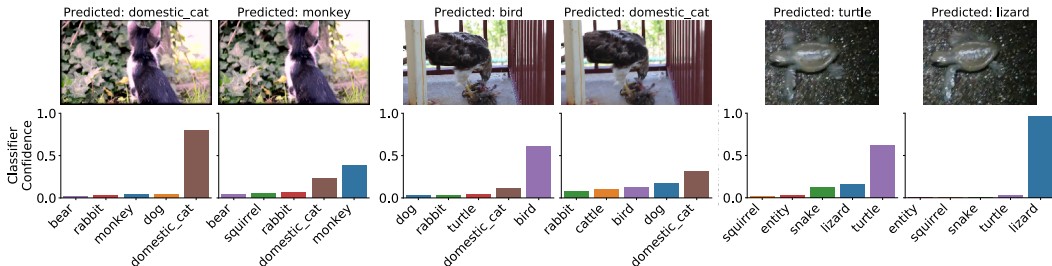

Figure 1: Three examples of natural perturbations from nearby video frames and resulting classifier confidences from a ResNet-152 model fine-tuned on ImageNet-Vid. While the images appear almost identical to the human eye, the classifier confidence changes substantially.

## 2  CONSTRUCTING A TEST SET FOR ROBUSTNESS

`ImageNet-Vid-Robust` and `YTBB-Robust` are sourced from videos in the ImageNet-Vid and Youtube-BB datasets (Russakovsky et al., 2015; Real et al., 2017). All object classes in ImageNet-Vid and Youtube-BB are from the WordNet hierarchy (Miller, 1995) and direct ancestors of ILSVRC-2012 classes. Using the WordNet hierarchy, we construct a canonical mapping from ILSVRC-2012 classes to ImageNet-Vid and Youtube-BB classes, which allows us to evaluate off-the-shelf ILSVRC-2012 models on `ImageNet-Vid-Robust` and `YTBB-Robust`. We provide more background on the source datsets in Appendix A.

### 2.1  CONSTRUCTING IMAGENET-VID-ROBUST AND YTBB-ROBUST

Next, we describe how we extracted sets of naturally perturbed frames from ImageNet-Vid and Youtube-BB to create `ImageNet-Vid-Robust` and `YTBB-Robust`. A straightforward approach would be to select a set of anchor frames and use temporally adjacent frames in the video with the assumption that such frames contain only small perturbations from the anchor. However, as Fig. 2 illustrates, this assumption is frequently violated, especially due to fast camera or object motion.

Instead, we first collect *preliminary* datasets of natural perturbations following the same approach, and then manually review each of the frame sets. For each video, we randomly sample an anchor frame and take $k = 10$ frames before and after the anchor frame as candidate perturbation images.[2] This results in two datasets containing one anchor frame each from 3,139 videos, with approximately 20 candidate perturbation per anchor frame.[3]

---

[1] We only evaluated detection on `ImageNet-Vid-Robust` as bounding-box annotations in Youtube-BB were only at 1 frame-per-second and not dense enough for our evaluation.

[2] For `YTBB-Robust` we use a subset of the anchor frames used by Gu et al. (2019).

[3] Anchor frames near the start or end of the video may have less than 20 candidate frames.

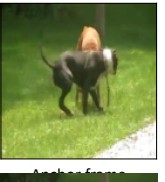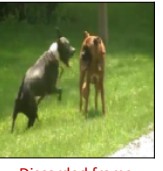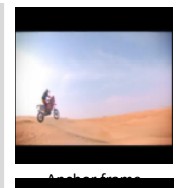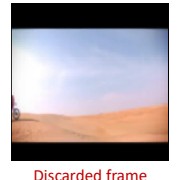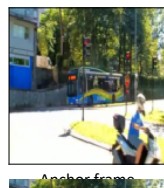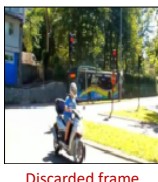

Anchor frame  Discarded frame  Anchor frame  Discarded frame  Anchor frame  Discarded frame

Figure 2: Temporally adjacent frames may not be visually similar. We show three randomly sampled frame pairs where the nearby frame was marked as "dissimilar" to the anchor frame during human review and then discarded from our dataset.

|  |  | ImageNet-Vid-Robust | YTBB-Robust |
|---|---|---|---|
| Anchor frames | Reviewed | 1,314 | 2,467 |
|  | Accepted | 1,109 (84%) | 2,030 (82%) |
|  | Labels updated | - | 834 (41%) |
| Frame pairs | Reviewed | 26,029 | 45,631 |
|  | Accepted | 21,070 (80.9%) | 36,827 (80.7%) |

Table 1: Statistics of ImageNet-Vid-Robust and YTBB-Robust. For YTBB-Robust, we updated the labels from for 41% (834) of the accepted anchors due to labeling errors in Youtube-BB.

Next, we curate the dataset with the help of four expert human annotators. The goal of the curation step is to ensure that each anchor frame and its nearby frames are correctly labeled with the same ground truth class, and that the anchor frame and the nearby frames are visually similar.

**Denser labels for Youtube-BB.** As Youtube-BB contains only a single category label per frame at 1 frame per second, annotators first viewed each anchor frame individually and marked any missing labels. In total, annotators corrected the labels for 834 frames, adding an average of 0.5 labels per anchor frame. These labels are then propagated to nearby, unlabeled frames at the native frame rate and verified in the next step. ImageNet-Vid densely labels all classes per frame, so we skip this step.

**Frame pairs review.** Next, for each pair of anchor and candidate perturbation frames, a human annotates (i) whether the pair is correctly labeled in the dataset, and (ii) whether the pair is similar. We took several steps to mitigate the subjectivity of this task and ensure high annotation quality. First, we trained reviewers to mark frames as dissimilar if the scene undergoes any of the following transformations: significant motion, significant background change, or significant blur change. We asked reviewers to mark each dissimilar frame with one of these transformations, or "other", and to mark a pair of images as dissimilar if a distinctive feature of the object is only visible in one of the two frames (such as the face of a dog). If an annotator was unsure about the correct label, she could mark the pair as "unsure". Second, we present only a single pair of frames at a time to reviewers because presenting videos or groups of frames could cause them to miss large changes due to the phenomenon of change blindness (Pashler, 1988).

**Verification.** In the previous stage, all annotators were given identical labeling instructions and individually reviewed a total of 71,660 images pairs. To increase consistency in annotation, annotators jointly reviewed all frames marked as dissimilar, incorrectly labeled, or "unsure". A frame was only considered similar to its anchor if a strict majority of the annotators marked the pair as such.

After the reviewing was complete, we discarded all anchor frames and candidate perturbations that annotators marked as dissimilar or incorrectly labeled. The final datasets contain a combined total of 3,139 anchor frames with a median of 20 similar frames each.

## 2.2 THE PM-K EVALUATION METRIC

Given the datasets introduced above, we propose a metric to measure a model's robustness to natural perturbations. In particular, let $A = \{a_1, ..., a_n\}$ be the set of valid anchor frames in our dataset. Let $Y = \{y_1, ..., y_n\}$ be the set of labels for $A$. We let $\mathcal{N}_k(a_i)$ be the set of frames marked as similar to anchor frame $a_i$. In our setting, $\mathcal{N}_k$ is a subset of the $2k$ temporally adjacent frames (plus/minus k frames from the anchor).

**Classification.** Classification accuracy is defined as $\text{acc}_{\text{orig}} = 1 - \frac{1}{N}\sum_{i=0}^{N}\mathcal{L}_{0/1}(f(a_i), y_i)$, where $\mathcal{L}_{0/1}$ is the standard 0-1 loss function. We define the `pm-k` analog of accuracy as

$$\text{acc}_{\text{pmk}} = 1 - \frac{1}{N}\sum_{i=0}^{N}\max_{b\in\mathcal{N}_k(a_i)}\mathcal{L}_{0/1}(f(b), y_i)\,, \tag{1}$$

which corresponds to picking the worst frame from each set $\mathcal{N}_k(a_i)$ before computing accuracy.

**Detection.** The standard metric for detection is mean average precision (mAP) of the predictions at a fixed intersection-over-union (IoU) threshold Lin et al. (2014). We define the `pm-k` metric analogous to that for classification: We replace each anchor frame with the nearest frame that minimizes the average precision (AP, averaged over recall thresholds) of the predictions, and compute `pm-k` as the mAP on these worst-case neighboring frames.

## 3 MAIN RESULTS

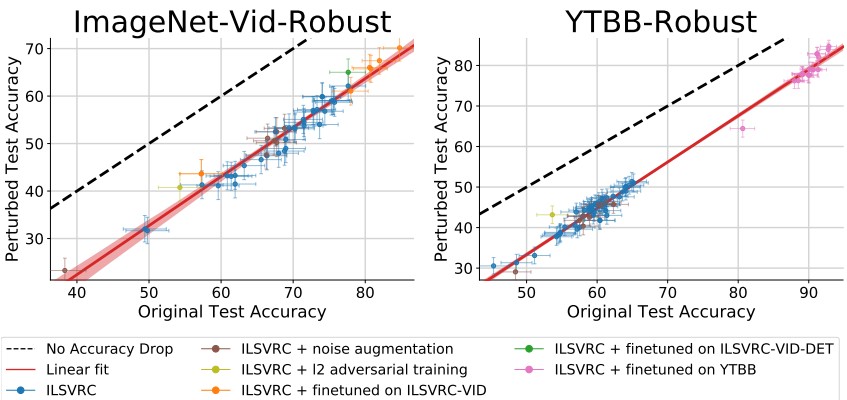

Figure 3: Model accuracy on original vs. perturbed images. Each data point corresponds to one model in our testbed (shown with 95% Clopper-Pearson confidence intervals). Each perturbed frame was taken from a ten frame neighborhood of the original frame (approximately 0.3 seconds). All frames were reviewed by humans to confirm visual similarity to the original frames.

We evaluate a testbed of 45 classification and three detection models on `ImageNet-Vid-Robust` and `YTBB-Robust`. We first discuss the various types of classification models evaluated with the `pm-k` classification metric. Second, we evaluate the performance of detection models on `ImageNet-Vid-Robust` using use the bounding box annotations inherited from ImageNet-Vid using a variant of `pm-k` for detection. We then analyze the errors made on the detection adversarial examples to isolate the effects of *localization* errors vs. *classification* errors.

### 3.1 CLASSIFICATION

The classification robustness metric is $\text{acc}_{\text{pmk}}$ defined in Equation (1). For frames with multiple labels, we count a prediction as correct if the model predicts *any* of the correct classes for a frame. In Figure 3, we plot the benign accuracy, $\text{acc}_{\text{orig}}$, versus the robust accuracy, $\text{acc}_{\text{pmk}}$, for all classification models in our test bed and find that the relationship between $\text{acc}_{\text{orig}}$ and $\text{acc}_{\text{pmk}}$ is approximately linear. This relationship indicates that improvements in the benign accuracy do result in improvements in the worst-case accuracy, but do not suffice to resolve the accuracy drop due to natural perturbations.

Our test bed consists of five model types with increasing levels of supervision. We present results for representative models from each model type in Table 2 and defer the full classification results table to Appendix B.2.

Table 2: Accuracies of five different model types and the best performing model. The model architecture is ResNet-50 unless noted otherwise. 'FT' is 'fine-tuning.' See Section 3.1 for details.

| Model Type | Accuracy Original | Accuracy Perturbed | $\Delta$ |
|---|---|---|---|
| `ImageNet-Vid-Robust` | | | |
| Trained on ILSVRC | 67.5 [64.7, 70.3] | 52.5 [49.5, 55.5] | 15.0 |
| + Noise Augmentation | 68.8 [66.0, 71.5] | 53.2 [50.2, 56.2] | 15.6 |
| + $\ell_\infty$ robustness (ResNext-101) | 54.3 [51.3, 57.2] | 40.8 [39.0, 43.7] | 12.4 |
| + FT on ImageNet-Vid | 80.8 [78.3, 83.1] | 65.7 [62.9, 68.5] | 15.1 |
| + FT on ImageNet-Vid (ResNet-152) | 84.8 [82.5, 86.8] | 70.2 [67.4, 72.8] | 14.6 |
| + FT on ImageNet-Vid-Det | 77.6 [75.1, 80.0] | 65.4 [62.5, 68.1] | 12.3 |
| `YTBB-Robust` | | | |
| Trained on ILSVRC | 57.0 [54.9, 59.2] | 43.8 [41.7, 46.0] | 13.2 |
| + Noise Augmentation | 62.3 [60.2, 64.4] | 45.7 [43.5, 47.9] | 16.6 |
| + $\ell_\infty$ robustness (ResNext-101) | 53.6 [51.4, 55.8] | 43.2 [41.0, 45.3] | 10.4 |
| + FT on Youtube-BB | 91.4 [90.1, 92.6] | 82.0 [80.3, 83.7] | 9.4 |
| + FT on Youtube-BB (ResNet-152) | 92.9 [91.6, 93.9] | 84.7 [83.0, 86.2] | 8.2 |

**ILSVRC Trained**    The WordNet hierarchy enables us to repurpose models trained for the 1,000 class ILSVRC dataset on `ImageNet-Vid-Robust` and `YTBB-Robust` (see Appendix A.1). We evaluate a wide array of ILSVRC-2012 models (available from Cadene) against our natural perturbations. Since these datasets present a substantial distribution shift from the original ILSVRC-2012 validation, we expect the *benign* accuracy acc$_{orig}$ to be lower than the comparable accuracy on the ILSVRC-2012 validation set. However, our main interest here is in the *difference* between the original and perturbed accuracies acc$_{orig}$ - acc$_{pmk}$. A small drop in accuracy would indicate that the model is robust to small changes that occur naturally in videos. Instead, we find significant drops of 15.0% and 13.2% in accuracy on our two datasets, indicating sensitivity to such changes.

**Noise augmentation**    One hypothesis for the accuracy drop from original to perturbed accuracy is that subtle artifacts and corruptions introduced by video compression schemes could degrade performance when evaluating on these corrupted frames. The worst-case nature of the `pm-k` metric could then be focusing on these corrupted frames. One model for these corruptions are the perturbations introduced in Hendrycks and Dietterich (2019). To test this hypothesis, we evaluate models augmented with a subset of the perturbations (exactly one of: Gaussian noise, Gaussian blur, shot noise, contrast change, impulse noise, or JPEG compression). We found that these augmentation schemes did not improve robustness against our perturbations substantially, and still result in accuracy drop of 15.6% and 16.6% on the two datasets.

$\ell_\infty$ **robustness.**    We evaluate the model from Xie et al. (2018), which currently performs best against $\ell_\infty$ attacks on ImageNet. We find that this model has a smaller accuracy drop than the two aforementioned model types on both datasets. However, we note that the robust model achieves *significantly* lower original and perturbed accuracy than either of the two model types above, and the robustness gain is modest (3% compared to models of similar benign accuracy).

**Fine-tuning on video frames.**    To adapt to the new class vocabulary and the video domain, we fine-tune several network architectures on the ImageNet-Vid and Youtube-BB training sets. For Youtube-BB, we train on the anchor frames used for training in Gu et al. (2019), and for ImageNet-Vid we use all frames in the training set. We provide hyperparameters for all models in Appendix K.

The resulting models significantly improve in accuracy over their ILSVRC pre-trained counterparts (e.g., 13% on `ImageNet-Vid-Robust` and 34% on `YTBB-Robust` for ResNet-50). This improvement in accuracy results in a modest improvement in the accuracy drop for `YTBB-Robust`, but a finetuned ResNet-50 still suffers from a significant 9.4% drop. On `ImageNet-Vid-Robust`, there is almost no change in the accuracy drop from 15.0% to 15.1%.

**Fine-tuning for detection on video frames.**    We further analyze whether additional supervision in the form of bounding box annotations improves robustness. To this end, we train the Faster R-CNN *detection* model Ren et al. (2015) with a ResNet-50 backbone on ImageNet-Vid. Following standard practice, the detection backbone is pre-trained on ILSVRC-2012. To evaluate this detector

for classification, we assign the class with the most confident bounding box as label to the image. We find that this transformation reduces accuracy compared to the model trained for classification (77.6% vs. 80.8%). While there is a slight reduction in the accuracy drop caused by natural perturbations, the reduction is well within the error bars for this test set.

## 3.2 DETECTION

We further study the impact of natural perturbations on object detection. Specifically, we report results for two related tasks: object localization and detection. Object detection is the standard computer vision task of correctly classifying an object and finding the coordinates of a tight bounding box containing the object. "Object localization", meanwhile, refers to only the subtask of finding the bounding box, *without* attempting to correctly classify the object.

We present our results on `ImageNet-Vid-Robust`, which contains dense bounding box labels unlike Youtube-BB, which only labels boxes at 1 frame per second. We use the popular Faster R-CNN Ren et al. (2015) and R-FCN Dai et al. (2016); Xiao and Jae Lee (2018) architectures for object detection and localization and report results in Table 3. For the R-FCN architecture, we use the model from Xiao and Jae Lee (2018)[4]. We first note the significant drop in mAP of 12 – 15 points for object detection due to perturbed frames for both the Faster R-CNN and R-FCN architectures. Next, we show that localization is indeed easier than detection, as the mAP is higher for localization than for detection (e.g., 76.6 vs 62.8 for Faster R-CNN with a ResNet-50 backbone). Perhaps surprisingly, however, switching to the localization task does *not* improve the drop between original and perturbed frames, indicating that natural perturbations induce both classification and localization errors. We show examples of detection failures in Figure 4.

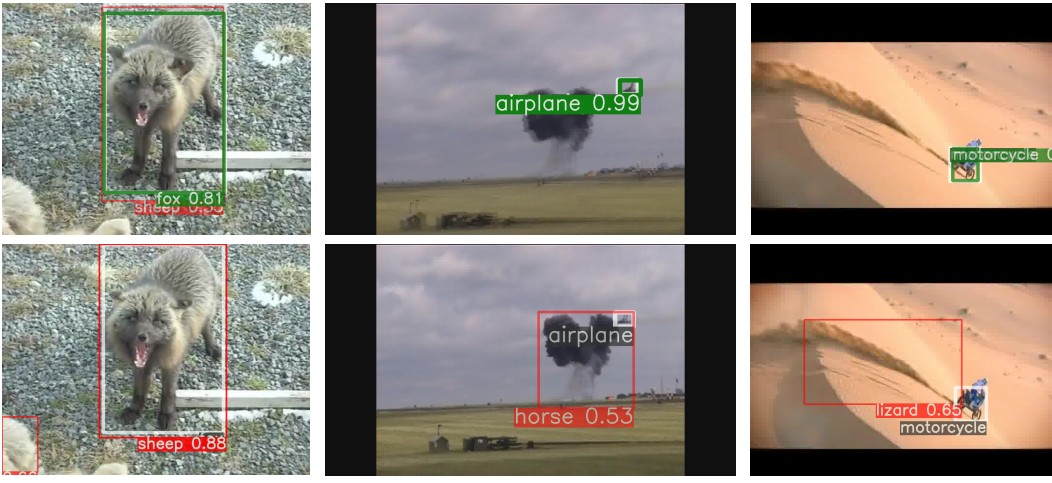

Figure 4: Naturally perturbed examples for detection. Red boxes indicate false positives; green boxes indicate true positives; white boxes are ground truth. Classification errors are common failures, such as the fox on the left, which is classified correctly in the anchor frame, and misclassified as a sheep in a nearby frame. However, detection models also have *localization* errors, where the object of interest is not correctly localized in addition to being misclassified, such as the airplane (middle) and the motorcycle (right). All visualizations show predictions with confidence greater than 0.5.

## 3.3 IMPACT OF DATASET REVIEW

We analyze the impact of our human review, described in Section 2.1, on the classifiers in our test bed. First, we compare the original and perturbed accuracies of a representative classifier (ResNet-152 finetuned) with and without review in Table 4. Our review improves the original accuracy by 3-4% by throwing away mislabeled or blurry anchor frames, and improves perturbed accuracy by 5-6% by discarding pairs of dissimilar frames. Our review reduces the accuracy drop by 1.8% on

---

[4]This model was originally trained on the 2015 subset of ImageNet-Vid. We evaluated this model on the 2015 validation set because the method requires access to pre-computed bounding box proposals which are available only for the 2015 subset of ImageNet-Vid.

Table 3: Detection and localization mAP for two Faster R-CNN backbones. Both detection and localization suffer from significant drops in mAP due to the perturbations. (*Model trained on ILSVRC Det and VID 2015 datasets, and evaluated on the 2015 subset of ILSVRC-VID 2017.)

| Task | Model | mAP Original | mAP Perturbed | mAP Δ |
|---|---|---|---|---|
| | FRCNN, ResNet 50 | 62.8 | 48.8 | 14.0 |
| Detection | FRCNN, ResNet 101 | 63.1 | 50.6 | 12.5 |
| | R-FCN, ResNet 101 Xiao and Jae Lee (2018)* | 79.4* | 63.7* | 15.7* |
| | FRCNN, ResNet 50 | 76.6 | 64.2 | 12.4 |
| Localization | FRCNN, ResNet 101 | 77.8 | 66.3 | 11.5 |
| | R-FCN, ResNet 101* | 80.9* | 70.3* | 10.6* |

`ImageNet-Vid-Robust` and 1.1% on `YTBB-Robust`, but still results in large accuracy drops. These results indicate that the changes in model predictions are indeed due to a lack of robustness, rather than due to significant differences between adjacent frames.

To further analyze the impact of our review on model errors, we plot how frequently each offset distance from the anchor frame results in a model error across all model types in Figure 5. For both datasets, larger offsets (indicating pairs of frames further apart in time) lead to more frequent model errors. Our review reduces the fraction of errors across offsets, and especially for large offsets, which are more likely to display large changes from the anchor frame.

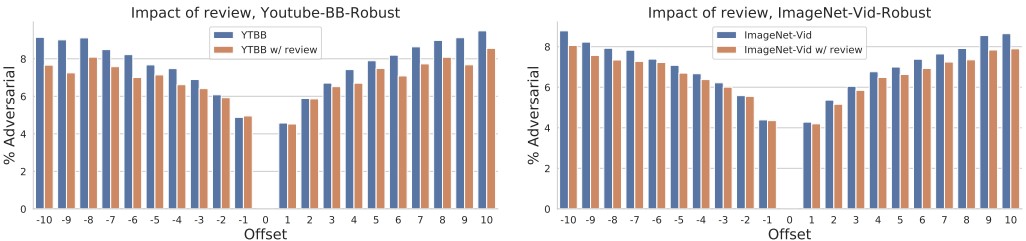

Figure 5: We plot the fraction of times each offset caused an error, across all evaluated models, for frames with and without review. Frames further away more frequently cause classifiers to misfire. Our review process reduces the number of errors, especially for frames further in time, by removing dissimilar frames.

Table 4: Impact of human review on original and perturbed accuracies for `ImageNet-Vid-Robust` and `YTBB-Robust`, using a ResNet-152 fine-tuned on ImageNet-Vid and Youtube-BB, respectively.

| | | Accuracy | | |
|---|---|---|---|---|
| | Reviewed | Original | Perturbed | Drop |
| ImageNet-Vid-Robust | ✗ | 80.3 | 64.1 | 16.2 |
| | ✓ | 84.8 | 70.2 | 14.4 |
| YTBB-Robust | ✗ | 88.1 | 78.1 | 10.0 |
| | ✓ | 92.9 | 84.7 | 8.9 |

## 4 RELATED WORK

**Adversarial examples.** While various forms of adversarial examples have been studied, the majority of research focuses on $\ell_p$ robustness Goodfellow et al. (2014); Biggio and Roli (2018). However, it is unclear whether adversarial examples pose a problem for classifier robustness outside of a truly worst case context. It is an open question whether perfect robustness against a $\ell_p$ adversary will induce robustness to realistic image distortions such as those studied in this paper. Recent work has proposed more realistic image modifications such as small rotations & translations Engstrom et al.

(2017); Azulay and Weiss (2018); Fawzi and Frossard (2015); Kanbak et al. (2017), hue and color changes Hosseini and Poovendran (2018), image stylization Geirhos et al. (2018a) and synthetic image corruptions such as Gaussian blur and JPEG compression Hendrycks and Dietterich (2019); Geirhos et al. (2018b). Even though the above examples are more realistic than the $\ell_p$ model, they still synthetically modify the input images to generate perturbed versions. In contrast, our work performs no synthetic modification and instead uses images that naturally occur in videos.

**Utilizing videos to study robustness.** In work concurrent to ours, Gu et al. (2019) exploit the temporal structure in videos to study robustness. However, their experiments suggest a substantially smaller drop in classification accuracy. The primary reason for this is a less stringent metric used in Gu et al. (2019). By contrast, our "pm-k" metric is inspired by the "worst-of-k" metric used in prior work Engstrom et al. (2017), highlighting the sensitivity of models to natural perturbations. In Appendix E we study the differences between the two metrics in more detail. Furthermore, the lack of human review and the high label error-rate we discovered in Youtube-BB(Table 1) presents a troubling confounding factor that we resolve in our work.

**Distribution shift.** Small, benign changes in the test distribution are often referred to as *distribution shift*. Recht et al. (2019) explore this phenomenon by constructing new test sets for CIFAR-10 and ImageNet and observe performance drops for a large suite of models on the newly constructed test sets. Similar to our Figure 3, the relationship between original and new test set accuracy is also approximately linear. However, the images in their test set bear little visual similarity to images in the original test set, while all of our failure cases in `ImageNet-Vid-Robust` and `YTBB-Robust` are on perceptually similar images. In a similar vein of study, Torralba et al. (2011) studies distribution shift *across* different computer vision data sets such as Caltech-101, PASCAL, and ImageNet.

**Computer vision.** A common issue when applying image based models to videos is *flickering*, where object detectors spuriously produce false-positives or false-negatives in isolated frames or groups of frames. Jin et al. (2018) explicitly identify such failures and use a technique reminiscent of adversarially robust training to improve image-based models. A similar line of work focuses on improving object detection in videos as objects become occluded or move quickly Kang et al. (2017); Feichtenhofer et al. (2017); Zhu et al. (2017); Xiao and Jae Lee (2018). The focus in this line of work has generally been on improving object detection when objects transform in a way that makes recognition difficult from a single frame, such as fast motion or occlusion. In this work, we document a broader set of failure cases for image-based classifiers and detectors and show that failures occur when the neighboring frames are imperceptibly different.

## 5 Conclusion

Our study quantifies the sensitivity of image classifiers to naturally occuring temporal perturbations. We show that these perturbations can cause significant drops in accuracy for a wide range of models for both classification and detection. Our work on analyzing this failure mode opens multiple avenues for future research:

**Building more robust models.** Our `ImageNet-Vid-Robust` and `YTBB-Robust` datasets provide a standard measure for robustness that can be applied to any classification or detection model. In Table 2, we evaluated several commonly used models and found that all of them suffer from substantial accuracy drops due to natural perturbations. In particular, we found that model improvements with respect to artificial perturbations (such as image corruptions or $\ell_\infty$ adversaries) induce at best modest improvements in robustness. We hope that our standardized datasets and evaluation metric will enable future work to quantify improvements in natural robustness directly.

**Further natural perturbations.** Videos provide a straightforward method for collecting natural perturbations of images, admitting the study of realistic forms of robustness for machine learning methods. Other methods for generating these natural perturbations are likely to provide additional insights into model robustness. As an example, photo sharing websites contain a large number of near-duplicate images: pairs of images of the same scene captured at different times, viewpoints, or from a different camera Recht et al. (2019). More generally, devising similar, domain-specific strategies to collect, verify, and measure robustness to natural perturbations in domains such as natural language processing or speech recognition is a promising direction for future work.

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

# A    SOURCE DATASET OVERVIEW

## A.1    IMAGENET-VID

The 2015 ImageNet-Vid dataset is widely used for training video object detectors Han et al. (2016) as well as trackers Bertinetto et al. (2016). We chose to work with the 2017 ImageNet-Vid dataset because it is a superset of the 2015 dataset. In total, the 2017 ImageNet-Vid dataset consists of 1,181,113 training frames from 4,000 videos and 512,360 validation frames from 1,314 videos. The videos have frame rates ranging from 9 to 59 frames per second (fps), with a median fps of 29. The videos range from 0.44 to 96 seconds in duration with a median duration of 12 seconds. Each frame is annotated with labels indicating the presence or absence of 30 object classes and corresponding bounding boxes for any label present in the frame. The 30 classes are ancestors of 293 of the 1,000 ILSVRC-2012 classes.

## A.2    YOUTUBE-BB

The 2017 Youtube-BB is a a large scale dataset with 8,146,143 annotated training frames 253,569 unique videos and with 1,013,246 validation frames from 31,829 videos. The video segments are approximately 19 seconds long on average. Each frame is annotated with exactly one label indicating the presence of 22 object classes, all of which are ancestors of 229 out of the ILSVRC-2012 classes.

# B    FULL ORIGINAL VS PERTURBED ACCURACIES

## B.1    IMAGENET-VID-ROBUST

| Model | Accuracy Original | Accuracy Perturbed | $\Delta$ |
|---|---|---|---|
| resnet152_finetuned | 84.8 [82.5, 86.8] | 70.2 [67.4, 72.8] | 14.6 |
| resnet50_finetuned | 80.8 [78.3, 83.1] | 65.7 [62.9, 68.5] | 15.1 |
| vgg16bn_finetuned | 78.0 [75.4, 80.4] | 61.0 [58.1, 63.9] | 17.0 |
| nasnetalarge_imagenet_pretrained | 77.6 [75.1, 80.1] | 62.1 [59.2, 65.0] | 15.5 |
| resnet50_detection | 77.6 [75.1, 80.1] | 65.0 [62.1, 67.8] | 12.6 |
| inceptionresnetv2_imagenet_pretrained | 75.7 [73.1, 78.2] | 58.7 [55.7, 61.6] | 17.0 |
| dpn107_imagenet_pretrained | 75.6 [72.9, 78.1] | 59.1 [56.1, 62.0] | 16.5 |
| inceptionv4_imagenet_pretrained | 75.3 [72.6, 77.8] | 59.0 [56.0, 61.9] | 16.3 |
| dpn92_imagenet_pretrained | 74.4 [71.7, 76.9] | 56.8 [53.8, 59.7] | 17.6 |
| dpn131_imagenet_pretrained | 74.0 [71.3, 76.6] | 59.9 [56.9, 62.8] | 14.1 |
| dpn68b_imagenet_pretrained | 73.7 [71.0, 76.2] | 54.0 [51.0, 57.0] | 19.7 |
| resnext101_32x4d_imagenet_pretrained | 73.3 [70.6, 75.9] | 57.2 [54.2, 60.1] | 16.1 |
| resnext101_64x4d_imagenet_pretrained | 72.9 [70.1, 75.5] | 56.6 [53.7, 59.6] | 16.3 |
| resnet152_imagenet_pretrained | 72.8 [70.0, 75.4] | 57.0 [54.0, 59.9] | 15.8 |
| resnet101_imagenet_pretrained | 71.5 [68.7, 74.1] | 53.7 [50.8, 56.7] | 17.8 |
| fbresnet152_imagenet_pretrained | 71.5 [68.7, 74.1] | 54.5 [51.5, 57.4] | 17.0 |
| densenet161_imagenet_pretrained | 71.4 [68.7, 74.1] | 55.1 [52.1, 58.1] | 16.3 |
| densenet169_imagenet_pretrained | 70.2 [67.5, 72.9] | 53.1 [50.1, 56.1] | 17.1 |
| densenet201_imagenet_pretrained | 70.2 [67.5, 72.9] | 53.4 [50.4, 56.4] | 16.8 |
| dpn68_imagenet_pretrained | 69.4 [66.6, 72.1] | 53.3 [50.3, 56.3] | 16.1 |
| bninception_imagenet_pretrained | 69.0 [66.2, 71.7] | 49.0 [46.0, 51.9] | 20.0 |
| densenet121_imagenet_pretrained | 69.0 [66.2, 71.7] | 50.9 [47.9, 53.8] | 18.1 |
| nasnetamobile_imagenet_pretrained | 68.8 [66.0, 71.5] | 48.4 [45.4, 51.4] | 20.4 |
| resnet50_augment___jpeg_compression | 68.8 [66.0, 71.5] | 53.2 [50.2, 56.2] | 15.6 |
| resnet34_imagenet_pretrained | 68.0 [65.2, 70.7] | 48.0 [45.0, 51.0] | 20.0 |
| resnet50_augment___impulse_noise | 67.7 [64.9, 70.5] | 50.2 [47.2, 53.2] | 17.5 |
| resnet50_augment__gaussian_blur | 67.7 [64.9, 70.5] | 52.5 [49.5, 55.5] | 15.2 |
| resnet50_imagenet_pretrained | 67.5 [64.7, 70.3] | 52.5 [49.5, 55.5] | 15.0 |
| resnet50_augment___gaussian_noise | 67.4 [64.5, 70.1] | 50.6 [47.6, 53.6] | 16.8 |
| resnet50_augment___shot_noise | 66.5 [63.6, 69.2] | 51.1 [48.1, 54.1] | 15.4 |
| vgg16_bn_imagenet_pretrained | 66.4 [63.5, 69.1] | 47.4 [44.5, 50.4] | 19.0 |
| resnet50_augment___defocus_blur | 66.3 [63.4, 69.1] | 47.6 [44.6, 50.6] | 18.7 |
| vgg19_bn_imagenet_pretrained | 65.6 [62.7, 68.4] | 46.6 [43.6, 49.6] | 19.0 |

| | | | |
|---|---|---|---|
| vgg19_imagenet_pretrained | 63.2 [60.3, 66.1] | 45.4 [42.4, 48.3] | 17.8 |
| resnet18_imagenet_pretrained | 61.9 [59.0, 64.8] | 41.5 [38.6, 44.4] | 20.4 |
| vgg13_bn_imagenet_pretrained | 61.9 [59.0, 64.8] | 43.3 [40.3, 46.3] | 18.6 |
| vgg16_imagenet_pretrained | 61.4 [58.5, 64.3] | 43.1 [40.2, 46.1] | 18.3 |
| vgg11_bn_imagenet_pretrained | 60.9 [57.9, 63.8] | 43.2 [40.3, 46.2] | 17.7 |
| vgg13_imagenet_pretrained | 59.6 [56.6, 62.5] | 41.1 [38.2, 44.1] | 18.5 |
| vgg11_imagenet_pretrained | 57.3 [54.4, 60.3] | 41.3 [38.4, 44.3] | 16.0 |
| alexnet_finetuned | 57.3 [54.3, 60.2] | 43.6 [40.7, 46.6] | 13.7 |
| ResNeXtDenoiseAll-101_robust_pgd | 54.3 [51.3, 57.2] | 40.8 [37.8, 43.7] | 13.5 |
| squeezenet1_1_imagenet_pretrained | 49.8 [46.8, 52.8] | 31.7 [28.9, 34.5] | 18.1 |
| alexnet_imagenet_pretrained | 49.4 [46.4, 52.4] | 32.0 [29.3, 34.8] | 17.4 |
| resnet50_augment___contrast_change | 38.3 [35.5, 41.3] | 23.3 [20.8, 25.9] | 15.0 |

Table 5: Classification model perturbed and original accuracies for all models in our test bed evaluated on the ImageNet-Vid-Robust dataset.

## B.2 YTBB–Robust

| Model | Accuracy Original | Accuracy Perturbed | Δ |
|---|---|---|---|
| resnet152_finetuned | 92.9 [91.2, 94.3] | 84.7 [82.4, 86.8] | 8.2 |
| resnet50_finetuned | 91.4 [89.6, 93.0] | 82.0 [79.6, 84.2] | 9.4 |
| inceptionresnetv2_finetuned | 91.3 [89.5, 92.9] | 79.0 [76.4, 81.3] | 12.3 |
| vgg19_finetuned | 90.5 [88.6, 92.2] | 79.1 [76.5, 81.4] | 11.4 |
| vgg16_finetuned | 89.1 [87.1, 90.8] | 78.0 [75.4, 80.4] | 11.1 |
| inceptionv4_finetuned | 88.5 [86.5, 90.3] | 76.3 [73.6, 78.7] | 12.2 |
| resnet18_finetuned | 88.0 [85.9, 89.8] | 76.2 [73.6, 78.7] | 11.8 |
| alexnet_finetuned | 80.6 [78.2, 82.9] | 64.4 [61.5, 67.3] | 16.2 |
| pnasnet5large_imagenet_pretrained | 65.2 [62.3, 68.0] | 51.0 [48.0, 54.0] | 14.2 |
| nasnetalarge_imagenet_pretrained | 64.9 [62.0, 67.7] | 51.4 [48.4, 54.4] | 13.5 |
| inceptionresnetv2_imagenet_pretrained | 64.5 [61.6, 67.4] | 50.4 [47.5, 53.4] | 14.1 |
| dpn98_imagenet_pretrained | 64.1 [61.2, 66.9] | 49.0 [46.0, 52.0] | 15.1 |
| dpn107_imagenet_pretrained | 64.1 [61.2, 66.9] | 50.1 [47.2, 53.1] | 14.0 |
| dpn131_imagenet_pretrained | 64.0 [61.1, 66.8] | 49.9 [46.9, 52.9] | 14.1 |
| inceptionv4_imagenet_pretrained | 63.6 [60.7, 66.4] | 48.8 [45.8, 51.8] | 14.8 |
| xception_imagenet_pretrained | 63.2 [60.2, 66.0] | 47.6 [44.6, 50.6] | 15.6 |
| dpn92_imagenet_pretrained | 62.3 [59.3, 65.1] | 47.7 [44.8, 50.7] | 14.6 |
| resnet50_augment__jpeg_compressioon | 62.3 [59.4, 65.2] | 45.7 [42.8, 48.7] | 16.6 |
| polynet_imagenet_pretrained | 61.4 [58.4, 64.3] | 47.3 [44.4, 50.3] | 14.1 |
| nasnetamobile_imagenet_pretrained | 61.4 [58.4, 64.3] | 43.0 [40.1, 46.0] | 18.4 |
| resnet50_augment__shot_noise | 61.3 [58.3, 64.2] | 46.4 [43.4, 49.3] | 14.9 |
| dpn68_imagenet_pretrained | 61.2 [58.3, 64.1] | 44.2 [41.2, 47.2] | 17.0 |
| fbresnet152_imagenet_pretrained | 61.1 [58.1, 64.0] | 45.9 [42.9, 48.8] | 15.2 |
| resnet152_imagenet_pretrained | 60.8 [57.8, 63.7] | 46.5 [43.5, 49.5] | 14.3 |
| resnet101_imagenet_pretrained | 60.8 [57.8, 63.7] | 45.2 [42.2, 48.2] | 15.6 |
| senet154_imagenet_pretrained | 60.7 [57.7, 63.6] | 47.2 [44.3, 50.2] | 13.5 |
| resnet50_augment__impulse_noise | 60.6 [57.7, 63.5] | 45.5 [42.6, 48.5] | 15.1 |
| se_resnet101_imagenet_pretrained | 60.5 [57.6, 63.4] | 45.6 [42.6, 48.6] | 14.9 |
| bninception_imagenet_pretrained | 60.4 [57.4, 63.3] | 41.8 [38.9, 44.7] | 18.6 |
| densenet161_imagenet_pretrained | 60.2 [57.3, 63.1] | 46.4 [43.4, 49.4] | 13.8 |
| resnet50_augment__gaussian_noise | 60.2 [57.3, 63.1] | 45.7 [42.8, 48.7] | 14.5 |
| se_resnext50_32x4d_imagenet_pretrained | 59.9 [56.9, 62.8] | 45.7 [42.7, 48.6] | 14.2 |
| dpn68b_imagenet_pretrained | 59.7 [56.7, 62.6] | 45.9 [42.9, 48.8] | 13.8 |
| inceptionv3_imagenet_pretrained | 59.6 [56.6, 62.5] | 43.8 [40.8, 46.8] | 15.8 |
| densenet121_imagenet_pretrained | 59.5 [56.5, 62.4] | 43.1 [40.1, 46.0] | 16.4 |
| se_resnext101_32x4d_imagenet_pretrained | 59.2 [56.3, 62.1] | 45.2 [42.3, 48.2] | 14.0 |
| densenet201_imagenet_pretrained | 59.2 [56.2, 62.1] | 44.8 [41.8, 47.8] | 14.4 |
| densenet169_imagenet_pretrained | 59.2 [56.2, 62.1] | 44.6 [41.7, 47.6] | 14.6 |

| | | |
|---|---|---|
| resnet50_augment__brightness_change | 58.9 [56.0, 61.8] | 42.6 [39.6, 45.5] | 16.3 |
| se_resnet50_imagenet_pretrained | 58.8 [55.9, 61.7] | 44.1 [41.1, 47.1] | 14.7 |
| se_resnet152_imagenet_pretrained | 58.8 [55.9, 61.7] | 44.8 [41.9, 47.8] | 14.0 |
| cafferesnet101_imagenet_pretrained | 58.2 [55.2, 61.1] | 44.3 [41.3, 47.3] | 13.9 |
| resnet50_augment__regular | 58.0 [55.1, 61.0] | 42.9 [39.9, 45.8] | 15.1 |
| resnet34_imagenet_pretrained | 57.9 [55.0, 60.9] | 42.8 [39.8, 45.7] | 15.1 |
| vgg19_imagenet_pretrained | 57.5 [54.6, 60.5] | 40.1 [37.2, 43.1] | 17.4 |
| resnet50_augment__gaussian_blur | 57.5 [54.5, 60.4] | 41.8 [38.9, 44.7] | 15.7 |
| vgg16_bn_imagenet_pretrained | 57.2 [54.2, 60.1] | 39.6 [36.7, 42.6] | 17.6 |
| resnet50_imagenet_pretrained | 57.0 [54.1, 60.0] | 43.8 [40.9, 46.8] | 13.2 |
| vgg19_bn_imagenet_pretrained | 56.8 [53.9, 59.8] | 40.6 [37.7, 43.5] | 16.2 |
| vgg16_imagenet_pretrained | 55.4 [52.4, 58.4] | 40.1 [37.2, 43.1] | 15.3 |
| vgg13_bn_imagenet_pretrained | 54.8 [51.8, 57.7] | 38.6 [35.7, 41.6] | 16.2 |
| vgg11_bn_imagenet_pretrained | 54.8 [51.8, 57.7] | 38.8 [35.9, 41.8] | 16.0 |
| vgg11_imagenet_pretrained | 54.7 [51.7, 57.6] | 38.4 [35.5, 41.3] | 16.3 |
| resnet18_imagenet_pretrained | 54.4 [51.4, 57.4] | 38.1 [35.2, 41.0] | 16.3 |
| vgg13_imagenet_pretrained | 54.2 [51.3, 57.2] | 37.7 [34.9, 40.7] | 16.5 |
| ResNeXtDenoiseAll-101_robust_pgd | 53.6 [50.7, 56.6] | 43.2 [40.2, 46.1] | 10.4 |
| squeezenet1_0_imagenet_pretrained | 51.1 [48.1, 54.1] | 33.1 [30.3, 36.0] | 18.0 |
| squeezenet1_1_imagenet_pretrained | 48.6 [45.6, 51.6] | 31.3 [28.6, 34.2] | 17.3 |
| resnet50_augment__defocus_blur | 48.4 [45.4, 51.4] | 29.1 [26.4, 31.8] | 19.3 |
| alexnet_imagenet_pretrained | 45.3 [42.4, 48.3] | 30.5 [27.8, 33.3] | 14.8 |

Table 6: Classification model perturbed and original accuracies for all models in our test bed evaluated on the YTBB-robust dataset..

## C    MODEL INDEPENDENT DISTRIBUTION SHIFT

Though the distribution shift we induced in our study were *model dependent* because we found the worst neighbor frame *for each* model, we could study the same problem but impose a static set of perturbed frames across all models. In Figure 6 we study this static set of perturbations across all models and see a substantial (but smaller) drop in accuracy for both models. The static set of perturbations were chosen by choosing the neighbor frame that the largest number of models classified incorrectly.

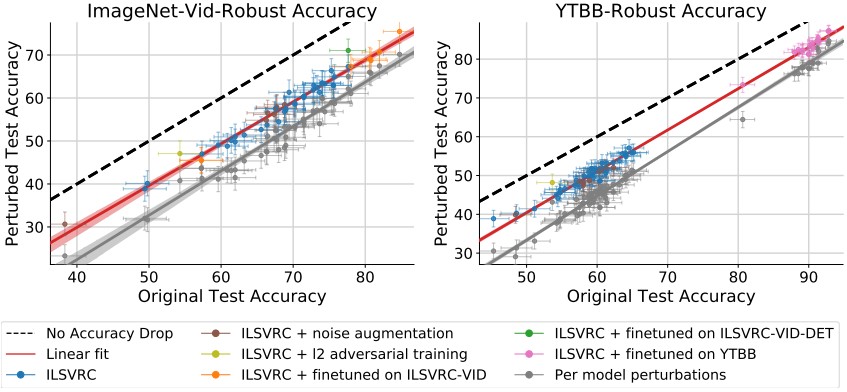

Figure 6: Model accuracy on original vs. perturbed images for a static set of perturbed frames across all models. The grey points and grey linear fit correspond to the perturbed accuracies of models evaluated on per model perturbations studied in Figure 3

## D    PER CLASS ANALYSIS

We study the effect of our perturbations on the 30 classes in `ImageNet-Vid-Robust` and `YTBB-Robust` to determine whether the performance drop was concentrated in a few "hard" classes.

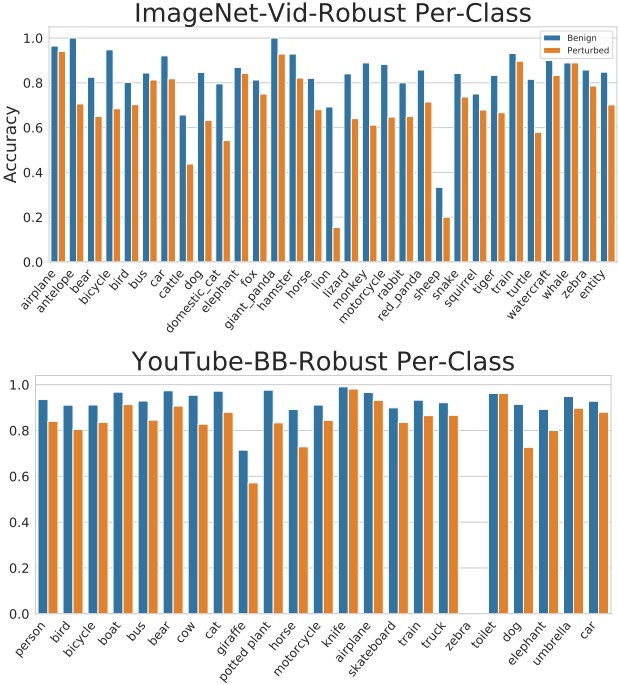

Figure 7: Per-class accuracy statistics for our best performing classification model (fine-tuned ResNet152) on `ImageNet-Vid-Robust` and `YTBB-Robust`. For Youtube-BB, note that 'zebra' is the least common label, present in only 24 anchor frames sampled by Gu et al. (2019), of which 4 are included in our dataset.

Figure 7 shows the original and perturbed accuracies across classes for our best performing model (a fine-tuned ResNet-152). Although there are a few particularly difficult classes for perturbed accuracy (e.g., lion or monkey on `ImageNet-Vid-Robust`), the accuracy drop is spread across most classes. On `ImageNet-Vid-Robust`, this model saw a total drop of 14.4% between original and perturbed images and a median drop of 14.0% in per-class accuracy. On `YTBB-Robust`, the total drop was 8.9% and the median drop was 6.7%.

## E  PER-FRAME CONDITIONAL ROBUSTNESS METRIC INTRODUCED IN GU ET AL. (2019)

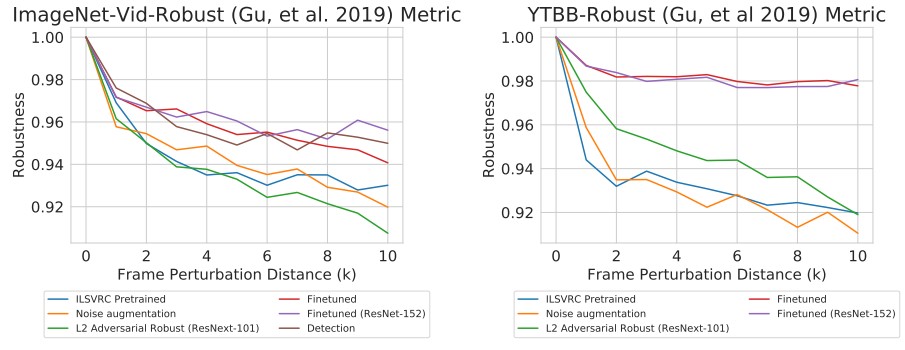

Figure 8: Conditional robustness metric from Gu et al. (2019) on perturbed frames as a function of perturbation distance on `ImageNet-Vid-Robust` and `YTBB-Robust`. Model accuracies from five different model types and the best performing model are shown. The model architecture is ResNet-50 unless otherwise mentioned.

In concurrent work, the authors of Gu et al. (2019) considered a different metric of robustness. In this section, we compute this metric on all models in our test bed to compare our findings to Gu et al. (2019). There are two main differences between PM-k and the robustness metric in Gu et al. (2019).

1. For two visually similar "neighbor" frames $I_0$ and $I_1$ with true label $Y$ and classifier $f$, Gu et al. (2019) studies the conditional probability $P(f(I_1) = y | f(I_0) = y)$
2. While PM-k looks for errors in all neighbor frames in a neighborhood of $k$ frames away from the anchor frame (so this would include frames 1, 2, ..., k frames away), Gu et al. (2019) only considers errors from **exactly** k frames away.

In Fig. 9 we illustrate simple example where two videos can have the same behavior for the metric introduced by Gu et al. (2019) but drastically different behavior for the PM-kmetric.

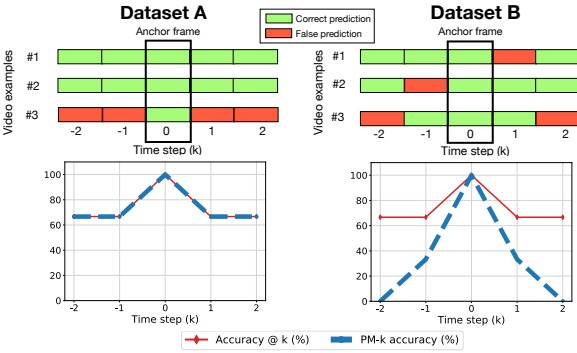

Figure 9: For the two example videos above the score from Gu et al. (2019) metric (Accuracy @ K) is identical, but the PM-k metric behaves substantially differently when the errors are spread across many independent videos, as shown in the right example

## F  $\ell_\infty$ DISTANCE VS PM-K ACCURACY

$\ell_\infty$ adversarial examples are well studied in the robustness community, yet the connection between $\ell_\infty$ and other forms of more "natural" robustness is unclear. Here, we plot the cumulative distribution of the $\ell_\infty$ distance between pairs of nearby frames in our datasets. In Figure 10, we show the CDF of $\ell_\infty$ distance for all pairs, all reviewed pairs, and mistakes made by 3 indicative models. Note the `fbrobust` model is trained specifically to be robust to $\ell_\infty$ adversaries.

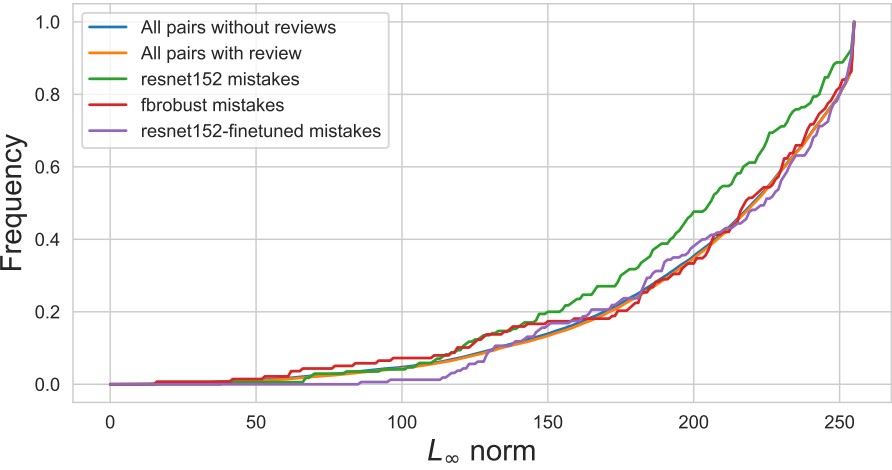

Figure 10: CDF showing the $\ell_\infty$ distance between pairs of frames from different distributions.

Table 7: Analyzing results based on frame-type in video compression. See Appendix H.1 for details.

|  | Original Acc. | Perturbed Acc. | $\Delta$ | # anchor frames |
|---|---|---|---|---|
| All frames | 84.8 | 70.2 | 14.6 | 1109 |
| w/o 'i-frames' | 84.7 | 70.3 | 14.4 | 1104 |
| w/o 'p-frames' | 83.9 | 73.7 | 10.2 | 415 |
| w/o 'b-frames' | 85.4 | 73.2 | 12.2 | 699 |

## G  PM-K ACCURACY WITH VARYING K

### G.1  IMAGENET-VID-ROBUST

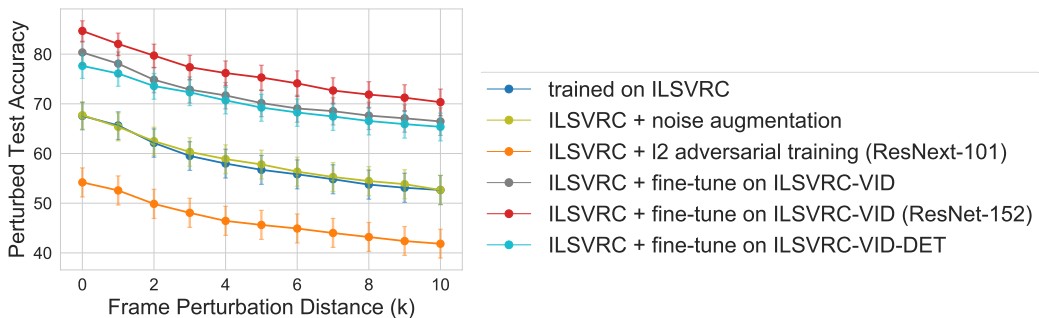

Figure 11: Model classification accuracy on perturbed frames as a function of perturbation distance (shown with 95% Clopper-Pearson confidence intervals). Model accuracies from five different model types and the best performing model are shown. The model architecture is ResNet-50 unless otherwise mentioned.

In Figure 11, we plot the relationship between $\text{acc}_{\text{pmk}}$ and perturbation distance (i.e., the `k` in the `pm-k` metric). The entire x-axis in Figure 11 corresponds to a temporal distance of at most 0.3 seconds between the original and perturbed frames.

## H  I-FRAMES AND P-FRAMES

### H.1  IMAGENET-VID-ROBUST

One possible concern with analyzing performance on video frames is the impact of video compression on model robustness. In particular, the videos in `ImageNet-Vid-Robust` contain 3 different frame types: 'i-frames', 'p-frames', and 'b-frames'. 'p-frames' are compressed by referencing pixel content from previous frames, while 'b-frames' are compressed via references to previous and future frames. 'i-frames' are stored without references to other frames.

We compute the original and perturbed accuracies, and the drop in accuracy for a subset of the dataset without 'i-frames', a subset without 'p-frames', and a subset without 'b-frames' in Table 7. While there are modest differences in accuracy due to compression, this analysis suggests that the sensitivity of models is not significantly due to the differences in quality of frames due to video compression.

## I  FPS ANALYSIS

### I.1  IMAGENET-VID-ROBUST

To analyze the impact of frame-rate on accuracy, we show results on subsets of videos with fixed fps (25, 29, and 30, which cover 89% of the dataset) using a fine-tuned ResNet-152 model in Table 8. The accuracy drop is similar across the subsets, and similar to the drop for the whole dataset.

| FPS | Acc. Orig. | Acc. Perturbed | Drop | # Videos |
|---|---|---|---|---|
| 25 | 87.3 [83.0, 90.9] | 73.3 [67.8, 78.3] | 14.0 | 292 |
| 29 | 87.7 [84.0, 90.8] | 74.9 [70.3, 79.2] | 12.8 | 383 |
| 30 | 78.3 [73.3, 82.7] | 61.7 [56.0, 67.1] | 16.6 | 313 |

Table 8: Results on subsets of ImageNet-Vid-Robust with fixed FPS.

## J ILSVRC TRAINING WITH IMAGENET−VID−ROBUST CLASSES

We trained ResNet-50 from scratch on ILSVRC using the 30 ImageNet-Vid classes. We also fine-tuned the model on ImageNet-Vid. In Table 9, we show the accuracy drops are consistent with models in our submission. We hypothesize that the lower accuracy is due to coarser supervision on ILSVRC.

| Model | Acc. Orig. | Acc. Perturbed | Drop |
|---|---|---|---|
| ILSVRC-30 | 61.0 | 44.9 | 15.1 |
| ILSVRC-30 + FT | 77.8 | 59.9 | 17.9 |

Table 9: Results of training ResNet-50 on ILSVRC with 30 classes from `ImageNet-Vid-Robust`.

## K EXPERIMENTAL DETAILS & HYPERPARAMETERS

All classification experiments were carried out using PyTorch version 1.0.1 on an AWS p3.2xlarge with the NVIDIA V100 GPU. All pretrained models were downloaded from Cadene at commit hash `021d97897c9aa76ec759deff43d341c4fd45d7ba`. Evaluations in Table **??** all use the default settings for evaluation. The hyperparameters for the *fine-tuned* models are presented in Table 10. We searched for learning rates between $10^{-3}$ and $10^{-5}$ for all models.

We additionally detail hyperparameters for detection models in Table 11. Detection experiments were conducted with PyTorch version 1.0.1 on a machine with 4 Titan X GPUs, using the Mask R-CNN benchmark repositoryMassa and Girshick (2018). We used the default learning rate provided in Massa and Girshick (2018). For R-FCN, we used the model trained by Xiao and Jae Lee (2018).

Table 10: Hyperparameters for models finetuned on ImageNet-Vid,

| Model | Base Learning Rate | Learning Rate Schedule | Batch Size | Epochs |
|---|---|---|---|---|
| resnet152 | $10^{-4}$ | Reduce LR On Plateau | 32 | 10 |
| resnet50 | $10^{-4}$ | Reduce LR On Plateau | 32 | 10 |
| alexnet | $10^{-5}$ | Reduce LR On Plateau | 32 | 10 |
| vgg16 | $10^{-5}$ | Reduce LR On Plateau | 32 | 10 |

Table 11: Hyperparameters for detection models.

| Model | Base Learning Rate | Learning Rate Schedule | Batch Size | Iterations |
|---|---|---|---|---|
| F-RCNN ResNet-50 | $10^{-2}$ | Step 20k, 30k | 8 | 40k |
| F-RCNN ResNet-101 | $10^{-2}$ | Step 20k, 30k | 8 | 40k |

## L DETECTION PM-K

We briefly introduce the mAP metric for detection here and refer the reader to Lin et al. for further details. The standard detection metric proceeds by first determining whether each predicted bounding box in an image is a true or false positive, based on the intersection over union (IoU) of the predicted and ground truth bounding boxes. The metric then computes the per-category average precision (AP, averaged over recall thresholds) of the predictions across all images. The final metric is reported as the mean of these per-category APs (mAP).

We define the `pm-k` analog of mAP by replacing each anchor frame in the dataset with a nearby frame that minimizes the per-image average precision. Since the category-specific average precision is undefined for categories not present in an image, we minimize the average precision across categories present in each frame rather than the mAP.

