# OpenReview forum: "Do Image Classifiers Generalize Across Time?"
_ICLR.cc/2020/Conference — Reject_

### Official Review · AnonReviewer3 · 2019-10-22
**Official Blind Review #3**

**Rating:** 3

**Review:**

This paper presents new datasets based on ImageNet and Youtube-BB to assert networks performance consistency across time. Compared to previous work, it uses human labeler to further validate the dataset and discard frames that are deemed too different from the reference one. It provides results on image classification and detection using popular network architectures. Based on these results, the paper claims an accuracy drop of 10 to 16%.

The main contribution of this paper is to introduce a new, human annotated dataset for robustness assessment of image classifiers. In itself, it is valuable work, but it is not clear if the contribution is important enough for ICLR. However, I would still be ok with accepting the paper (better datasets are always useful) if it was not for the way the results are reported. I do not specifically have issues with "more stringent robustness metric" but it should not be used to claim incredible results (like an accuracy drop of 10 to 16% instead of 3% for previous work (Real et al. 2017)).

There is one thing for sure: using "accuracy drop" in this context is just misleading. The underlying concept to which "accuracy" refers is _not_ the "maximum error made by the network over the whole set of images". By this definition of accuracy, if the number of images around the reference frame were 100, missing a _single one_ each time (that is 99% of actual accuracy) would result, according to this peculiar redefinition, to a _0%_ accuracy. This is actually highlighted in Appendix G.1: the "accuracy" trend can only go down, since every supplemental frame brings one more chance to fail and obtain a 0% accuracy for this set of perturbed images.
Same thing goes for the detection, where the only frame that matters among all is the one _minimizing_ the AP. Same thing in Table 4, where the "accuracy" of the Original column means one thing (the amount of correctly identified images over the total number of images) while the "accuracy" of the Perturbed column right next to it means something completely different. Same thing in Table 2, which even provides a "delta" between two unrelated metrics.
I cannot see how this can be justified. Sure, there could be some usage for such strict metric, but again, this is _not_ accuracy and cannot be compared to any previous results. Having a more stringent metric is one thing, but in this case it just seems like a justification to get high drop numbers.

Keeping that in mind, these are the actual conclusions we can make from the paper:
1) Human reviewers removed or changed about 20% of the frames
2) This resulted in a relative accuracy improvement of about 4% for the reference frame (Table 4, column Original). The improvement for the perturbed frames are not actually provided.
3) The comparison (and improvements) to previous work due to the dataset cleaning remains unclear.
4) Comparison between different networks and training procedures

Overall, the paper presents impressive numbers but does not actually back them up. I am open to eventually consider acceptance given the value of the datasets, but the paper would then require a significant overhaul to remove all confusing aspects.

**Experience Assessment:**

I have read many papers in this area.

**Review Assessment: Checking Correctness Of Derivations And Theory:**

N/A

**Review Assessment: Checking Correctness Of Experiments:**

I assessed the sensibility of the experiments.

**Review Assessment: Thoroughness In Paper Reading:**

I read the paper thoroughly.

---

> ### Author Response · Authors · 2019-11-09
> **Response to reviewer 3**
>
> We understand R3’s concern that calling the worst-case pm-k metric an “accuracy drop” might be misleading. However, we argue that the PM-k “accuracy drop” is analogous to the “adversarial accuracy drop” claimed in the adversarial example and robustness literature [Athalye et al (2018) https://arxiv.org/pdf/1802.00420.pdf,  Madry et al. (2017) https://arxiv.org/pdf/1706.06083.pdf]. When reporting adversarial accuracy of a classifier on a dataset, one reports the maximum error made in a \delta ball around each image; in this work we are reporting the maximum error over a set of naturally perturbed images temporally and visually close to the anchor frame. The K in PM-K behaves analogously to \delta in L_p adversaries in that the adversarial accuracy is monotonically decreasing in \delta.
>
> Unfortunately, we cannot follow R3's comment that "[human review] resulted in a relative accuracy improvement of about 4% for the reference frame (Table 4, column Original). The improvement for the perturbed frames are not actually provided." In Table 4, we do show the improvement on the perturbed frames in the column labeled "Perturbed", and show that human review results in a 5-6% improvement on ImageNet-Vid-Robust and YTBB-Robust.

---

> > ### Comment · AnonReviewer3 · 2019-11-14
> > **Accuracy is accuracy**
> >
> > > However, we argue that the PM-k “accuracy drop” is analogous to the “adversarial accuracy drop” claimed in the adversarial example and robustness literature
> >
> > Adversarial papers do not "reports the maximum error made in a \delta ball around each image": there is no exhaustive search, where one would feed the network with all possible images at \delta distance. Each adversarial method returns one and only one image, which is estimated to be the worst case error in this \delta radius. The accuracy is then computed as: (number of images for which the network got the right answer) / (number of image given to the network)
> > In other words, the canonical definition of accuracy.
> >
> > Nonetheless, an illustrative parallel can be drawn with adversarial attacks. Suppose that one invent a new adversarial approach, which goes as follow:
> > 1. For each image in the dataset, generate 1,000,000,000 adversarial images inside the \delta radius, using various existing methods and adding a bit of randomness to obtain slightly different perturbed image each time. If the network misclassifies a _single_ one of these 1,000,000,000 tries, then we consider it to have failed.
> > 2. Compute the accuracy as M/N, where N is the number of images in the dataset and M the number of images for which the network correctly classified every single one of the 1B associated adversarial examples.
> >
> > Such methodology would definitely obtain a quite spectacular drop in "accuracy" compared to the state of the art. Is it still meaningful to talk about accuracy in this context, though? I do not think so, and it is the same thing here.
> >
> > The second sentence of my point 2 above was indeed incorrect. Overall, the contributions of the paper are thus:
> > 1) A human validation of two existing datasets of video perturbations, which results in a 4% relative accuracy improvement for the reference frame and ~6% for the frames around.
> > 2) Results for different networks on a metric which cannot be compared to any other work.
> >
> > We add to this the misleading abstract wording and intro ("We evaluate a diverse array of classifiers pre-trained on ImageNet and show a median classification accuracy drop of 16 and 10 percent on our two datasets."). Realistically speaking, what would a reasonably knowledgeable reader think when reading the words "median accuracy drop of 16%":
> > A) wow, that's a significant drop compared to the mere 3% reported in previous work, they probably uncover something very important and up to now unseen, or
> > B) well, "median accuracy drop" probably means something different than in any textbook or previous work on the matter, nothing to see here.
> >
> > Sure, presenting the actual median accuracy drop would have resulted in less impressive numbers. And that's precisely my point.
> >
> > My opinion thus remains. The results presented in the paper are not plainly wrong, but the contributions are very slim yet worded in a way that makes them look huge. I do not see value in such an overstated paper.

---

> > > ### Author Response · Authors · 2019-11-15
> > > **Adversarial Accuracy is Worst Case accuracy in Lp ball.**
> > >
> > >  > “Adversarial papers do not "reports the maximum error made in a \delta ball around each image": there is no exhaustive search, where one would feed the network with all possible images at \delta distance. Each adversarial method returns one and only one image, which is estimated to be the worst case error in this \delta radius. The accuracy is then computed as: (number of images for which the network got the right answer) / (number of image given to the network) In other words, the canonical definition of accuracy.”
> > >
> > > We respectfully disagree with this assertion made by reviewer 3.
> > >
> > > The definition of adversarial robustness involves the maximum possible error in a perturbation set. For Lp adversaries, this would be the existence of a misclassified example within a delta ball *irrespective* of how many examples within that delta ball are tested.  A perfect attack would by definition have a higher error rate than the 1e9 example attack mentioned by the reviewer, and be considered a valid attack giving a score of 0% on that example if a single adversarial example exists in the delta ball that is misclassified.
> > >
> > > With regards to the experiment Reviewer 3 outlined. We refer the reviewer to https://arxiv.org/abs/1902.02322.  In the evaluation section, Carlini performs an almost identical experiment as the reviewer suggested and shows that the attack is 100% successful (Carlini only generates 25 examples not 1e9). This method is seen by the community as a valid way to attack models. This technique is also related to random-restarts, another commonly used approach in generating adversarial examples. Here, noise is added to the input image and a search procedure such as PGD is started from each instantiation of the noise (see Madry et al, 2017 https://arxiv.org/pdf/1706.06083.pdf, Section 3.1).
> > >
> > >
> > > Furthermore, the predominant technique to generate adversarial examples, Projected Gradient Descent, often searches through *many* examples before finding an incorrectly classified adversarial example. In fact, Carlini et al’s monograph “On Evaluating Adversarial Robustness” (https://arxiv.org/pdf/1902.06705.pdf)  mentions in Section 4.8:
> > >
> > >  " ```For example, on CIFAR-10 or ImageNet with a maximum ℓ∞ distortion of 8/255, white-box optimization attacks generally converge in under 100 or 1000 iterations with a step size of 1. However, black-box attacks often take orders of magnitude more queries, with attacks requiring over 100,000 queries not being abnormal. For a different dataset, or a different distortion metric, attacks will require a different number of iterations. While it is possible to perform an attack correctly with very few iterations of gradient descent, it requires much more thought and care (Engstrom et al., 2018). For example, in some cases even a million iterations of white-box gradient descent have been found necessary (Qian & Wegman, 2018).```"
> > >
> > > Each step of PGD generates a potential adversarial example (a different image), and up 1e6 iterations are necessary in some cases. If we measure accuracy as reviewer 3 suggests, an adversarial example that takes 1e6 PGD iterations to generate had (1e6 - 1) correctly classified examples and 1 incorrect example, but the adversarial accuracy on this example is reported as 0% not 99.9999% (note all (1e6 - 1) correctly classified examples are in fact within the delta ball).
> > >
> > > We thus stand by the definition of “accuracy drop” used in our paper.

---

### Official Review · AnonReviewer2 · 2019-10-22
**Official Blind Review #2**

**Rating:** 3

**Review:**

This paper targets on the evaluation of model robustness on similar video frames. The authors build two carefully labeled video datasets, and extensive experiments are conducted to show that the state-of-the-art classification and detection models are not robust enough when dealing with very similar video frames.  The results are similar to my intuitive feelings.

The authors propose acc_orig (the average acc) and acc_pmk (which chooses the worse one in nearest 2k frames) to amplify the gap. I personally think acc_pmk is too stringent. I wonder if there is still large gaps if we choose a random frame in the nearest 2k frames.

The authors have tried fine-tuning and data augmentation techniques to improve the robustness, although the performance is improved, the gap between acc_orig and acc_pmk does not change much.

The paper has done many work to analyze the robustness of image classification and detection models, however, the results are expected and no effective methods are proposed to improve the results. Overall, the contribution is limited.

**Experience Assessment:**

I have read many papers in this area.

**Review Assessment: Checking Correctness Of Derivations And Theory:**

I assessed the sensibility of the derivations and theory.

**Review Assessment: Checking Correctness Of Experiments:**

I assessed the sensibility of the experiments.

**Review Assessment: Thoroughness In Paper Reading:**

I read the paper at least twice and used my best judgement in assessing the paper.

---

> ### Author Response · Authors · 2019-11-09
> **Response to Reviewer 2**
>
> We thank the reviewer for their comments.
>
> Regarding the stringency of our metric: the worst-case nature of our metric is inspired by adversarial robustness, where worst-case notions are commonplace. Moreover, adversarial robustness is more stringent than our metric since it allows *any* l_p bounded perturbation, not only images occurring naturally in videos. The goal of our paper is to find a middle ground between standard accuracy (which does not measure robustness) and l_p adversarial robustness. We refer the reviewer to our more detailed response to R3 regarding the metric.
>
> Further, while we are happy to hear that the results match R2's expectations, we argue that systematically measuring robustness is critical for progress in this direction. Especially for the question of robustness, it is crucial to rigorously quantify various phenomena. We believe a key contribution of our work is in verifying the intuitions of R2 (and possibly others in the field).
>
> Upon R2's request, we compute the accuracy drop on a random frame in the nearest 20 frames for the best performing models on the two datasets (a fine-tuned Resnet-152 in both cases).
>
> For Youtube-BB-robust the anchor accuracy was 92.8%, the PM-K accuracy was 84.6%, and the random neighbor frame accuracy was 93.1%.
>
> For ImageNet-Vid-Robust, anchor accuracy was 84.8%, PM-K accuracy was 70.1%, and a random neighbor frame accuracy was 84.1%.
>
> The accuracy differences between the anchor frame accuracy and sampling a random frame are within error bars.  This is to be expected: Our anchor frames are randomly sampled, so sampling random frames within a neighborhood of the anchor frames is analogous to a new, iid draw of anchor frames, resulting in little change in accuracy. The adversarial nature of our PM-K metric is the reason for the large drops.

---

### Official Review · AnonReviewer1 · 2019-10-31
**Official Blind Review #1**

**Rating:** 6

**Review:**

Summary
In this paper, the authors curated two datasets: ImageNet-Vid and Youtube-BB in order to create human-reviewed perceptibly similar sets (Imagenet-Vid-Robust and YTBB-Robust). The obtained datasets are evaluated over 45 different models pre-trained on ImageNet in order to see their drop in accuracy on natural perturbations. Three detection models are also evaluated and show that not only classification models are sensitive to these perturbations, but that it also yields to localization errors.

Comments
The paper is clear, well organized, well written and easy to follow.
The authors present two novel datasets grouped in sets of perceptibly similar images and answer to the following hypothesis: Can the perturbations occurring naturally in videos be a realistic robustness challenge?
The thorough evaluation over the curated datasets shows pretty well that the changes in the model prediction are indeed due to a lack of robustness of the models themselves rather than the difference occurring from one frame to the other (occlusion etc).
The authors mention the curation was done with the help of expert human annotators. Details could be added as to how the annotators are considered experts and what process they went through (mturk? Handmade application to select the frames?).
Overall I think the paper adds an interesting contribution, with the datasets themselves which can be used for image similarity tasks for example
Although the contribution of the paper is important, it seems limited for the conference with no novel method proposed.

Typo
Section 3, l 4: using use -> using


**Experience Assessment:**

I do not know much about this area.

**Review Assessment: Checking Correctness Of Derivations And Theory:**

I assessed the sensibility of the derivations and theory.

**Review Assessment: Checking Correctness Of Experiments:**

I assessed the sensibility of the experiments.

**Review Assessment: Thoroughness In Paper Reading:**

I read the paper at least twice and used my best judgement in assessing the paper.

---

> ### Author Response · Authors · 2019-11-09
> **Response to Reviewer 1**
>
> We thank the reviewer for their detailed comments and are happy to hear that they found the paper well organized with a thorough evaluation.
>
> Regarding details about the curation process: The expert annotators in this case were the authors of the paper. We designed a custom interface for annotating frames, which displayed the anchor frame next to a single frame that is nearby in the video. For each pair, the annotators stated whether the nearby frame was similar or dissimilar to the anchor frame. We will add these details, along with a depiction of the interface, in the appendix in the next revision. A few screenshots of the UI can be found here: https://pictureweb.s3-us-west-2.amazonaws.com/iclr/ui.html

---

### Decision · Program_Chairs · 2019-12-19

**Decision:**

Reject

**Comment:**

This paper proposed to evaluate the robustness of CNN models on similar video frames. The authors construct two carefully labeled video databases. Based on extensive experiments, they conclude that the state of the art classification and detection models are not robust when testing on very similar video frames. While Reviewer #1 is overall positive about this work, Reviewer #2 and #3 rated weak reject with various concerns. Reviewer #2 concerns limited contribution since the results are similar to our intuition. Reviewer #3 appreciates the value of the databases, but concerns that the defined metrics make the contribution look huge. The authors and Reviewer #3 have in-depth discussion on the metric, and Reviewer #3 is not convinced. Given the concerns raised by the reviewers, the ACs agree that this paper can not be accepted at its current state.